# Virtual serious games for women's health education: A scoping review

**Krissy Jordan**[iD]\*, **Christine Kurtz Landy, Celina Da Silva, Mahdieh Dastjerdi, Bella Grunfeld**

School of Nursing, York University, Toronto, Ontario, Canada

\* kjordan@yorku.ca

## Abstract

### Background

Virtual serious games (VSGs) offer an engaging approach to women's health education. This review examines the state of research on VSGs, focusing on intended users, design characteristics, and assessed outcomes.

### Methods

Following JBI methodology guidance for the scoping review, searches were conducted in the MEDLINE, CINAHL, EMBASE, Web of Science, and PsycINFO databases from inception to April 22, 2024. Eligible sources included participants: women or females aged 18 years and older, with no restrictions based on health condition or treatment status; concept: VSGs; context: settings where health education is provided. Sources were restricted to English language and peer-reviewed articles. Two reviewers independently screened titles, abstracts, and full texts using eligibility criteria. Data extraction was performed by one reviewer and verified by another using a custom tool. Quantitative (e.g., frequency counting) and qualitative (content analysis) methods were employed. The findings were organized into figures and tables accompanied by a narrative description.

### Results

12 studies from 2008 to 2023, mostly in the U.S. (66.7%), explored various age groups and women's health, focusing on breast and gynecological cancer (67%). Half (50%) of the VSGs were theory-informed; 41.7% involved users, and 58.3% had partnerships. Game types included tablet (41.7%), mobile (25%), and web (33.3%). Gameplay dosage varied from single session (50%) to self-directed (25%) and specific frequency (25%). Gameplay duration was self-directed (50%) or fixed lengths (50%). Outcomes included knowledge (50%), skills (16.7%), satisfaction (58.3%), health-related metrics (41.7%), and gameplay analysis (16.7%).

**Data availability statement:** All relevant data are within the article and its Supporting Information files. The study protocol is available on protocols.io at DOI: https://dx.doi.org/10.17504/protocols.io.j8nlkd4z5g5r/v1

**Funding:** The author(s) received no specific funding for this work.

**Competing interests:** The authors have declared that no competing interests exist.

## Conclusions

Studies show increased interest in VSGs for women's health education, especially regarding breast and gynecological cancer. The focus on theoretical frameworks, user involvement, and collaborations highlights a multidisciplinary approach. Varied game modalities, dosage, and assessed outcomes underscore VSG adaptability. Future research should explore long-term effects of VSGs to advance women's health education.

## Introduction

Women's health education is crucial for informed decision-making. With comprehensive knowledge, women can better understand their bodies and overall well-being. This includes various health-related topics such as menstruation, pregnancy, contraception, and menopause. Access to accurate, trustworthy, and reliable information allows women to advocate for their health needs, navigate healthcare systems, and actively participate in decisions impacting their well-being.

To effectively address women's unique health needs, providing accessible and tailored education and information is important. However, traditional approaches to health education, such as brochures or didactic presentations, often lack interactivity throughout the educational experience. This gap in interactive learning methods warrants examination, especially considering studies revealing that many women lack the necessary information to make informed healthcare decisions, particularly in childbirth [1–5]. Additionally, research has shown that the clinician's attitudes, beliefs, and preferences influence their counselling [6,7], highlighting an important gap in care provision based on personal biases. Therefore, innovative approaches that effectively empower and educate women about their health in an unbiased way are necessary to bridge these information gaps and overcome these challenges.

One potential strategy for addressing the limitations of traditional approaches is using virtual simulations. Virtual simulations are "clinical simulations offered on a computer, the Internet, or in a digital learning environment, including single or multiuser platforms" [8, p. 27]. Virtual simulations can take various forms, such as computer-based scenarios, immersive virtual reality environments, and augmented experiences, including virtual serious games (VSGs), also known as serious games. While virtual simulations and VSGs are often used interchangeably, it is important to note that they are distinct concepts with different focuses. Simulations aim to replace real-world activities and environments, allowing learners to practice skills and apply knowledge in a controlled setting [9]. In contrast, VSGs are interactive digital games designed for educational purposes rather than pure entertainment [10,11]. Specifically, VSGs employ game design elements and structures for non-entertainment purposes, engaging players in activities that promote learning and skill acquisition [12]. VSGs incorporate educational content into gameplay and engage players through rules, rewards, and consequences, motivating them to accomplish specific activities and learning objectives [13,14].

Extensive research has demonstrated the successful use of VSGs in health professionals' education, reflected in enhanced user experiences and improved knowledge or skills [15–20]. However, despite the growing amount of research on VSGs in healthcare professionals' education, there is a notable absence in identifying and mapping existing literature specifically focused on VSGs tailored for women's health education. This gap is surprising, given the significant involvement of women in gaming, where they comprise 51% of the gaming community and dedicate an average of 7.9% hours per week to gameplay, as highlighted by the Entertainment Software Association of Canada (ESAC) [21].

A preliminary search of seminal databases including MEDLINE, the Cochrane Database of Systematic Reviews, and *JBI Evidence Synthesis* revealed the absence of systematic or scoping reviews addressing VSGs for women's health education. Considering the demonstrated advantages of VSGs in educating healthcare professionals, conducting a scoping review focusing on women's health education becomes imperative. In doing so, researchers can consolidate evidence on VSGs targeting women's education and identify important aspects of VSGs to inform future work in this field.

The primary objective of this scoping review is to identify and map the state of current research on VSGs designed for women's health education. This review will provide a detailed analysis of the specific populations of women targeted by VSGs, the characteristics (i.e., targeted areas of women's health, theoretical perspectives, engagement of women, collaborations and partnerships with developers, gaming modalities, and gameplay dosage) and the outcomes examined in these VSGs. Through this examination, we will identify gaps in the current research and offer valuable suggestions for future studies on VSGs for women's health education. By conducting this scoping review, we aim to contribute to advancing knowledge and the development of effective educational interventions for women's health.

### Review question(s)

What is the current state of research on VSGs designed for women's health education?
This is divided into the following sub-questions:

(i)  Which specific populations of women are targeted by VSGs in current research?

(ii)  What are the characteristics of VSGs designed for women's health education?

(iii)  What outcomes have been examined in studies that evaluated VSGs designed for women's health?

### Methods

This scoping review was conducted in accordance with the JBI methodology for scoping reviews [22] and in line with the Preferred Reporting Items for Systematic Reviews and Meta-Analyses extension for Scoping Reviews (PRISMA-ScR) [23]. An *a priori* protocol was developed to guide the review process. To promote transparency and reproducibility, the protocol was registered and made publicly available on protocols.io. It can be accessed via the following DOI: dx.doi.org/10.17504/protocols.io.j8nlkd4z5g5r/v1.

### Inclusion criteria

**Participants.**  This review considered studies exclusively involving participants identifying as women or females aged 18 years and older with no restrictions based on health condition or treatment status. Additionally, studies focused on educating healthcare professionals about women's health were not considered within the scope of this review.

**Concept.**  This review considered studies that explored VSGs intended for women's health education. In this context, VSGs are interactive digital games primarily designed for educational purposes rather than pure entertainment [10,11].

To maintain the emphasis on women's health education, studies focusing solely on entertainment or lacking an educational component related to women's health were excluded. Exergames, which aim to promote exercise [24], without providing health education, were excluded. In addition, studies utilizing games for therapeutic purposes, such as pain

management or anxiety reduction, without an educational component were excluded. Furthermore, studies solely focusing on gamification, defined as using game design elements (i.e., badges, leaderboards, points, evoked emotions, narratives, and competition) in traditionally nongame contexts [25,26], were excluded as gamification is distinct from VSGs. Gamification is a design technique applied to existing learning activities or curricula to achieve goals, while VSGs begin as a game designed to fulfill specific objectives [26].

**Context.** The context for this scoping review included any settings where health education is provided to women, such as healthcare facilities (hospitals, clinics, and health centres), community centres, schools and universities, workplaces, and other relevant institutions or organizations. Health education is "any combination of learning experiences designed to help individuals… improve their health by increasing knowledge, influencing motivation, and improving health literacy" [27, p. 18]. No limitations were set for the geographical locations of this review.

**Types of sources.** This scoping review considered quantitative, qualitative, and mixed methods study designs for inclusion. In addition, systematic reviews, and scoping reviews were considered for inclusion in this scoping review. Sources of grey literature, such as conference abstracts, opinion papers, dissertations, and other unpublished material, were excluded to ensure the inclusion of peer-reviewed studies that offered sufficient methodological rigour and detail for data extraction.

## Search strategy

The search strategy aimed to locate published primary studies and systematic or scoping reviews. An initial limited search in MEDLINE (Ovid) and CINHAL (EBSCO) was undertaken to identify articles on the topic. The text words in the titles and abstracts of relevant articles and the index terms used to describe the articles were used to develop a full search strategy. Input was sought from a research librarian during the search strategy developed for the MEDLINE application in other electronic databases. The search strategy, including all identified keywords and index terms, was adapted for each included information source and a final search was undertaken on April 22, 2024. The full search strategies are provided in S1 File. The reference lists of articles selected for full-text review were screened, and no additional articles were found.

Studies published in English were included to ensure accurate understanding and interpretation due to the authors' proficiency in the language of the reviewed material. The search was conducted without specific timeframe limitations to ensure the capture of all relevant studies related to our research questions.

The databases searched included MEDLINE (Ovid), CINHAL (EBSCO), EMBASE (Ovid), Web of Science (Clarivate), and PsycINFO (Ovid).

## Study/source of evidence selection

All identified citations and accompanying abstracts were collated and uploaded into EndNote Version 20 (Clarivate Analytics, PA, USA). The citations and abstracts were transferred into Covidence systematic review software (Veritas Health Innovation, Melbourne, Australia), and duplicates were removed. Following a pilot test, titles and abstracts were screened by two independent reviewers (KJ and BG) for assessment against the inclusion and exclusion criteria for the review. Potentially relevant sources were retrieved in full and imported into Covidence. Full-text sources that did not meet the inclusion criteria were excluded, and reasons for their exclusion are summarized in S2 File. Any disagreements that arose between the reviewers were resolved through discussion until consensus was established. Authors of papers were contacted to request missing or additional data, where required.

## Data extraction

The first author (KJ) extracted data using a custom data extraction tool developed by the research team to gather pertinent information from the articles (See S3 File). The data extracted included specific details about the participants, concept, context, and relevant findings addressing the review questions. The second reviewer (BG) verified the data by

reviewing the completed data extraction tool alongside the selected sources, as one approach recommended by Peters et al [22]. Any disagreements that arose between the reviewers were resolved through discussion until consensus was established.

Data items that were extracted included:

- Study details and characteristics (e.g., author, publication year, country, methodology, study design, method of data collection, sample size, and timing of knowledge post-test after completion of game);

- Population details (e.g., age, considerations of racial and ethnic groups);

- Details related to the characteristics of the VSGs relating to women's health education (i.e., targeted areas of women's health, theoretical framework used for VSG development, engagement of women from the target group, collaborations and partnerships in development, gaming modalities, and gameplay dosage);

- Information pertaining to the outcomes examined in studies that evaluated VSGs targeting women's health education.

### Data analysis and presentation

In this scoping review, we utilized conventional content analysis [28] to analyze the data. Specifically, we extracted codes directly from the data, focusing on specific words and phrases that described general study characteristics, population details, the characteristics of VSGs relevant to women's health education, and the outcomes examined within these groups. These codes were then sorted into categories and further subdivided within the data extraction table. This approach aligns with the JBI methodology for scoping reviews [22].

Additionally, we conducted descriptive statistical analysis, focusing on frequency counts and percentages, to assess various aspects, such as general study characteristics, population details, characteristics of the VSGs relevant to women's health education, and outcomes examined within the VSGs. The results are organized based on the scoping review's objectives and presented in tables. Along with the tabular presentations, a narrative summary addresses the research questions.

## Results

### Study inclusion

A total of 4198 titles and abstracts were identified from the database searches, and no additional references were identified through the screening of references. After removing duplicates, 3103 references were screened for eligibility by reviewing titles and abstracts, resulting in 37 references for full-text review. Full-text screening was conducted in duplicate by two independent reviewers, with discrepancies resolved through discussion. From this set, references were excluded based on ineligible participants (n = 10), ineligible interventions (n = 7), and ineligible sources (n = 8) as displayed in Fig 1. Twelve (n = 12) studies were included in this review [29–40] and are summarized in a PRISMA flow diagram [41] in Fig 1. Four studies examined the same VSG intervention [31,33,38,39]. One study was a pilot randomized clinical trial examining the intervention's feasibility, acceptability, and preliminary efficacy [39], another examined game engagement and its associated learning outcomes [38], a third provided information on the development and acceptability of the intervention [33], and the fourth provided a qualitative cross-sectional study describing women's perspectives using the intervention [31]. All four studies were included due to their relevance and ability to complement each other.

### Characteristics of included studies

All studies included in the review were published between 2008 and 2023, with eleven studies (91.7%) published in the last 10 years. Eight studies (66.7%) were from the United States [29–33,35,38,39], one (8.3%) from Nepal [34], one (8.3%) from South Korea [37], one (8.3%) from Malaysia [40], and one (8.3%) from Norway [36]. Almost all included

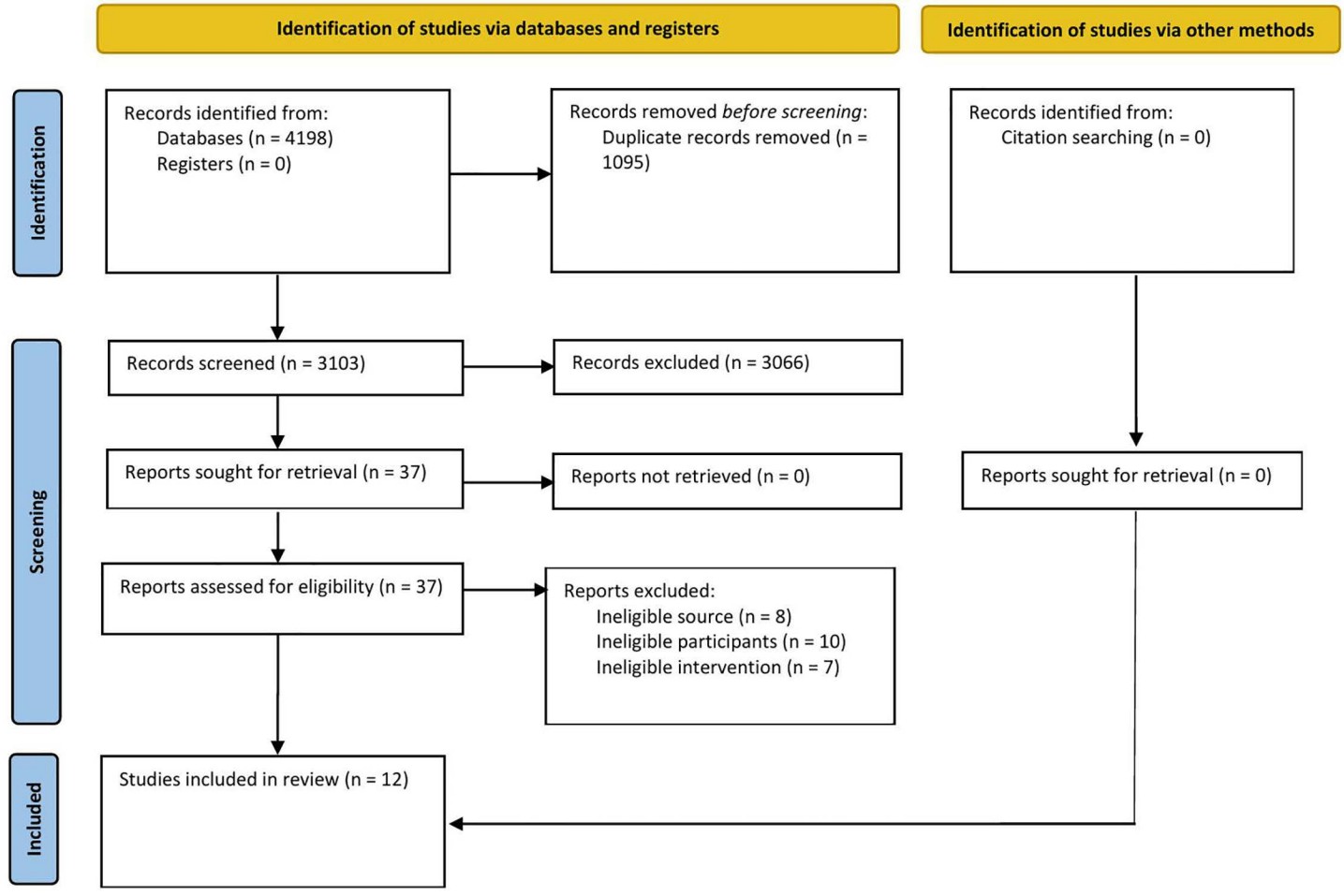

**Fig 1. Search results and study selection and inclusion process [41].**

studies were published as journal articles reporting quantitative (n = 8) [29,32,35–40], qualitative (n = 2) [31,34], and mixed methods research studies (n = 2) [30,33]. Sample sizes varied from 26 [29] to 4518 [36] participants in quantitative studies, 40 [31] to 61 [34] in qualitative studies, and 10 [33] to 34 [32] in mixed methods studies. Methods for data collection included focus groups [34], semi-structured interviews [31,33,34,37], surveys/questionnaires [32–35,37–40], pre-post knowledge assessment [29,34,38–40], registries [36], electronic healthcare record [39], gameplay data [32,38], self-reported measures [32], observation [33], field notes [31], and voice journals [31]. Knowledge post-test timings varied after VSG completion; four studies (33.3%) assessed immediately [29,30,34,40]. Another study (8.3%) had assessments at 1, 2, and 3-month intervals [32], while another one (8.3%) conducted tests at VSG completion and 10–12 days later [35]. Table 1 presents a detailed review of the included studies characteristics.

## Review findings

This section presents key findings addressing each of the research questions.

Table 1 presents the findings that answer question 1: What specific populations of women are targeted by VSGs in current research?

**Table 1. Characteristics of included studies.**

| Author(s) (Year of Publication) Country of Origin | Aims | Methodology/Study Design | Participants (Age and Sample Size) | Data Collection | Research Outcomes Assessed/Measured |
|---|---|---|---|---|---|
| Hill & Coker (2022) USA [29] | Would video logs (vlog) delivered as storytelling be more effective at increasing knowledge than a question-and-answer model? | Quantitative Prospective, Randomized controlled trial | Cisgender Black or African American women aged 18–45; *n* = 26 | Pre-post test assessments | • Changes in knowledge<br>• Reported behaviors |
| Kayastha *et al.* (2021) Nepal [34] | The research question and key objectives of this study are to understand the attitudes of women about mobile games, understand the acceptability and usability of the prototype MANTRA intervention by women and female community health volunteers (FCHVs) and to explore their perceptions of knowledge change brought about by the mHealth gaming application. | Qualitative | Women aged 20–60 years; *n* = 61 | Focus groups Interviews Pre-post knowledge assessment | • Perspectives<br>• Knowledge |
| Kim *et al.* (2018) South Korea [37] | The objective of our study was to evaluate if patient education using a mobile game may increase drug compliance, decrease physical side effects of chemotherapy, and improve psychological status in breast cancer patients. | Quantitative Randomized controlled trial | Women with metastatic breast cancer aged 18–65 years; *n* = 76 | Questionnaires | • Time spent on education<br>• Compliance to medication n<br>• Physical side effects<br>• Psychological side effects including quality of life<br>• Satisfaction in ILOVEBREAST |
| Kukafka *et al.* (2015) USA [30] | The purpose of this study is to conduct focus groups to inform the iterative design of RealRisks, a patient-centered decision aid for communicating breast cancer risk, reducing inaccurate risk perceptions, and providing preference-based decision support for risk management. | Mixed methods | Women aged 35–75 years; *n* = 34 | Focus groups Questionnaires | • Perceived breast cancer risk<br>• Acceptability of RealRisks |
| Orumaa *et al.* (2022) Norway [36] | This matched retrospective cohort study examined the association between exposure to the FightHPV mobile app gamified educational content and having a cervical exam in the following year. | Quantitative Retrospective cohort study | Women aged 20–69 years; *n* = 4518 | Registries | • Screening participation (i.e., having a cervical exam)<br>• Cervical exam result |
| Shiyko *et al.* (2016) USA [32] | The goal of this study was to evaluate real-world play patterns of SpaPlay and its impact on body mass index (BMI) and nutritional knowledge. | Quantitative 90-day longitudinal study | Women aged 21 years and older; *n* = 47 | Telemetry game play data Self-reported measures Questionnaire | • BMI<br>• Nutritional knowledge<br>• Telemetry game play data<br>• Readiness to change behaviors in domains of exercise, nutrition, and consumption of sweetened beverages |
| Silk *et al.* (2008) USA [35] | To examine the effectiveness of three modalities for delivery of nutrition education. | Quantitative Between-subjects, repeated-measures design | Low-income European American and African American mothers, aged 18–50 years; *n* = 155 | Surveys | • Likeability<br>• Nutrition knowledge |

*(Continued)*

**Table 1.** (Continued)

| Author(s) (Year of Publication) Country of Origin | Aims | Methodology/Study Design | Participants (Age and Sample Size) | Data Collection | Research Outcomes Assessed/Measured |
|---|---|---|---|---|---|
| Thomas *et al.* (2019) USA [33] | The purpose of this study was to develop and assess the initial acceptability of a serious game to teach women with advanced cancer self-advocacy skills, including communication, decision-making, and social connectivity, to improve their quality of life with cancer. | Mixed methods (a) an open pilot study of a draft version of the serious game, (b) development of the Strong Together serious game prototype, and (c) assessment of initial acceptability of the Strong Together prototype. | Women with advanced gynecological or metastatic breast cancer, aged 18 years or older; pilot *n* = 10, development *n* = 10; acceptability *n* = 15 | Observation Questionnaires Semi-structured interviews | • Initial acceptability |
| Thomas *et al.* (2023) USA [31] | The purpose of this study is to describe participant perspectives of a novel, self-advocacy serious game intervention called Strong Together. | Qualitative Cross-sectional descriptive study | Women with advanced gynecological or metastatic breast cancer, aged 18 years or older; *n* = 40 | Semi-structured interviews Field notes Voice journals | • Acceptability |
| Thomas et al. (2023) USA [39] | The aim of this pilot randomized clinical trial was to assess the feasibility, acceptability, and preliminary efficacy of a novel, low-intensity, real-world serious game intervention (Strong Together) compared to enhanced care as usual for women newly diagnosed with advanced breast or gynecology cancer. | Quantitative Pilot randomized clinical trial | Women with advanced gynecological or metastatic breast cancer, aged 18 years or older; *n* = 78 | Questionnaires/ surveys Exit interview Electronic medical records | • Acceptability<br>• Feasibility<br>• Preliminary efficacy on self-advocacy skills<br>• Health-related quality of life and symptom burden |
| Tong & Qi Hee (2023) Malaysia [40] | The purpose of this study is to design and evaluate the impact of an educational game on breast cancer awareness among female university students in Malaysia. | One-group pre-and post-intervention pilot study | Malaysian female from private and public higher education, aged 18 years or older; *n* = 52 | Questionnaire | • Changes in breast cancer awareness |
| You et al. (2023) USA [38] | This pilot study aims to describe game engagement and its associations with learning outcomes, sociodemographics, and health factors in women with advanced cancer receiving a 12-week self-advocacy serious game intervention. | Quantitative Pilot study | Women with advanced gynecological or metastatic breast cancer, aged 18 years or older; *n* = 38 | Survey Study tablets | • Game engagement<br>• Patient-reported outcomes (self-advocacy, quality of life, symptom burden, mood, and social and demographic information) |

**Study population.** The studies encompassed a diverse representation of women across various age groups. Participant age ranges varied across the studies, including women aged 18 years or older [30,31,33,38–40], specifically ranging from 18–45 years [29], 20–60 years [34], 18–65 years [37], 35–75 years [30], 20–69 years [36], those older than 21 years [32], and 18–50 years [35]. Additionally, three studies (25%) considered ethnic and racial diversity, targeting specific populations such as Black or African American women [29], low-income European, American, and African American mothers [35], and Malaysian women [40].

Table 2 presents the findings that answer question 2: What are the characteristics of VSGs designed for women's health education? The studies described several details, including target areas of women's health, theoretical perspectives, engagement of women from the target group in the VSG development process, collaboration and partnerships in VSG development among diverse stakeholders, gaming modalities, and gameplay dosage.

**Table 2. Characteristics of virtual serious games (VSGs).**

| Author(s) (Year of Publication) Country of Origin | Target Areas of Women's Health | Theoretical Framework | Involvement of Target Group | Collaboration and partnerships in VSG development | Gaming Modality | Frequency of Game | Duration of Game | Duration Between Game and Post-Test |
|---|---|---|---|---|---|---|---|---|
| Hill & Coker (2022) USA [29] | Human immunodeficiency virus (HIV) and sexually transmitted infections (STI) | Not discussed | Not discussed | Not discussed | Tablet-based | Single session, unclear repetition | 8 min 46 sec | Immediately |
| Kayastha et al. (2021) Nepal [34] | Maternal health | Not discussed | Community members, including Female Community Health Volunteers (FCHVs) | Health Research and Social Development Forum (HERD) | Mobile-based | Single session, unclear repetition | 10–30 min | Immediately |
| Kim et al. (2018) South Korea [37] | Breast cancer | Not discussed | Not discussed | Not discussed | Mobile-based | Recommend three times per week | 22.2 min Recommended 30 min per week | Not evaluated |
| Kukafka et al. (2015) USA [30] | Breast cancer | Not discussed | Not discussed | Not discussed | Web-based | Single session, unclear repetition | 30 min | Immediately |
| Orumaa et al. (2022) Norway [36] | Gynecological cancer | Not discussed | Not discussed | Not discussed | Mobile-based | Self-directed | Self-directed | Not evaluated |
| Shiyko et al. (2016) USA [32] | Nutrition | Self-determination theory Player experience of need satisfaction (PENS) model | Not discussed | Partnership between academia and industry over 5 years | Web-based | Self-directed | Self-directed | 1-, 2-, and 3- month intervals |
| Silk et al. (2008) USA [35] | Nutrition | Media uses and gratifications (MUG) theory | Not discussed | Collaboration between MSU Extension and MSU Communication Technology Laboratory | Web-based | Single session, unclear repetition | 20–30 min | Time 1 – immediately Time 2 – approx. 10–12 days after |
| Thomas et al. (2019) USA [33] | Breast or gynecological cancer | Female self-advocacy in cancer survivorship Behavior intervention technology (BIT) Framework | Co-design process with target population; considering patient's self-advocacy | Oncology clinicians, researchers, and advocacy organizations included the National Ovarian Cancer Coalition Simcoach Games, a Carnegie Mellon University-affiliated serious games educational technology firm | Tablet-based | Single session, unclear repetition | Average 19:13 minutes (range 12:15–27: 26 minutes) | Not evaluated |

*(Continued)*

 

**Table 2.** (Continued)

| Author(s) (Year of Publication) Country of Origin | Target Areas of Women's Health | Theoretical Framework | Involvement of Target Group | Collaboration and partnerships in VSG development | Gaming Modality | Frequency of Game | Duration of Game | Duration Between Game and Post-Test |
|---|---|---|---|---|---|---|---|---|
| Thomas *et al.* (2023) USA [31] | Breast or gynecological cancer | Female self-advocacy in cancer survivorship Behavior intervention technology (BIT) Framework | Co-design process with target population; considering patient's self-advocacy | Oncology clinicians, researchers, and advocacy organizations included the National Ovarian Cancer Coalition Simcoach Games, a Carnegie Mellon University-affiliated serious games educational technology firm | Tablet-based | Self-directed | Self-directed | Not evaluated |
| Thomas *et al.* (2023) USA [39] | Breast or gynecological cancer | Female self-advocacy in cancer survivorship Behavior intervention technology (BIT) Framework | Co-design process with target population; considering patient's self-advocacy | Oncology clinicians, researchers, and advocacy organizations included the National Ovarian Cancer Coalition Simcoach Games, a Carnegie Mellon University-affiliated serious games educational technology firm | Tablet-based | Suggested once a week & repeat session | Self-directed | Baseline, 3 and 6 months |
| Tong & Qi Hee (2023) Malaysia [40] | Breast cancer | Not discussed | Not discussed | Not discussed | Web-based | Single session, unclear repetition | Self-directed | Immediately |
| You *et al.* (2023) USA [38] | Breast or gynecological cancer | Female self-advocacy in cancer survivorship Behavior intervention technology (BIT) Framework | Co-design process with target population; considering patient's self-advocacy | Oncology clinicians, researchers, and advocacy organizations included the National Ovarian Cancer Coalition Simcoach Games, a Carnegie Mellon University-affiliated serious games educational technology firm | Tablet-based | Recommended once a week for 12 weeks and repeat game scenarios to explore various response options | Self-directed | Baseline, 3 and 6 months |

**Targeted areas of women's health.** The VSGs studies covered various areas of women's health education. Eight studies (67%) focused on breast or gynecological cancer [30,31,33,36–40]. Two studies (16.6%) focused on nutrition [32,35], one study (8.3%) was related to maternal health [34], and one study (8.3%) focused on human immunodeficiency virus (HIV) and sexually transmitted infections (STIs) [29].

**Theoretical perspectives.** Six studies (50%) described using theoretical perspectives [31–33,35,38,39]. Theoretical perspectives included the self-determination theory [32], the player experience of need satisfaction (PENS) model [32], the media uses and gratifications (MUG) theory [35], and the conceptual framework of female self-advocacy in cancer survivorship and the behavior intervention technology (BIT) framework [31,33,38,39].

**Engagement of women from the target group in the design of the VSG.** Seven (of 12) studies (58.3%) did not provide information on how they included or interacted with women from the target group when designing the VSG [29,30,32,35–37,40]. Conversely, five studies (41.7%) demonstrated some level of participation or interaction with women from the target group [31,33,34,38,39]. One study (8.3%) invited community members, including the Female and Community Health Volunteers (FCHVs), to participate to ensure the VSG's appropriateness and understanding [34]. The remaining four studies (33.3%) employed a multistage, user-centred co-design process with the target population to develop the game's content, taking into account patients' self-advocacy and preferences [31,33,38,39].

**Collaboration and partnerships in VSG development among diverse stakeholders.** Out of the 12 studies reviewed, the level of involvement and collaboration in VSG development varied across the interventions, and included academic researchers, industry partners, clinicians, and advocacy organizations. Five studies (41.7%) did not report collaborations or partnerships among these groups [29,30,36,37,40]. However, seven studies (58.3%) mentioned the involvement of collaborators or partners in the development process [31–35,38,39]. One study (8.3%) conducted the development process with researchers from the Health Research and Social Development Forum (HERD) [34]. An additional study (8.3%) highlighted a long-term partnership between academia and industry, spanning over five years [32], while another study reported collaboration between the Michigan State University (MSU) Extension and the MSU Communication Technology Laboratory [35]. Four studies (33.3%) discussed partnerships with oncology clinicians, researchers, and advocacy organizations, including the National Ovarian Cancer Coalition, and Simcoach Games, a Carnegie Mellon University-affiliated serious games educational technology firm [31,33,38,39].

**Gaming modality.** The gaming modalities used in the interventions varied across the studies. The most frequently used modality was tablet-based, and was implemented in five studies (41.7%) [29,31,33,38,39]. Mobile-based gaming was used in three studies (25%) [34,36,37]; web-based gaming was employed in four studies (33.3%), where interventions were accessed through personal computers or laptops [30,32,35,40].

**Gameplay dosage.** The results of the studies reviewed revealed varying levels of gameplay dosage (i.e., the amount of gameplay involved, including frequency and duration). In terms of gameplay frequency (i.e., the number of gameplay sessions), six studies (50%) permitted gameplay during a single session; however, it remained unclear if repeated gameplay occurred [29,30,33–35,40]. Three studies (25%) adopted a self-directed approach, where participants could play the game as little or as much as they desired [31,32,36]. In contrast, three studies (25%) recommended a specific gameplay frequency, suggesting participants play the game once a week with repeated sessions [38,39] or three times per week [37].

As for gameplay duration (i.e., the length of time interacting with the VSG), six studies (50%) adopted a self-directed approach, allowing participants to decide how long they played the VSG [31,32,36,38–40]. Conversely, six studies (50%) specified the length of gameplay [29,30,33–35,37]. The most commonly reported duration of gameplay was 30 minutes [30,34,35] while the shortest duration reported was 8 minutes and 46 seconds long [29].

To answer question 3, What outcomes have been examined in studies that evaluated VSGs designed for women's health, Table 1 presents the findings.

**Outcomes examined.** Six studies (50%) examined knowledge as an outcome [29,30,32,34,35,40], with a focus on nutritional knowledge (16.7%) [32,35], HIV and STIs (8.3%) [29], breast cancer risk or awareness (16.7%) [30,40], and maternal and child health knowledge related to disasters (8.3%) [34]. Two studies (16.7%) assessed self-advocacy skills [38,39]. Seven studies (58.3%) explored acceptance, likability, or satisfaction (58.3%) [30,31,33–35,37,39]. Additionally, health-related outcomes (41.7%) were investigated, including health-related behaviours (16.7%) [29,32], medication compliance, side effects of chemotherapy, quality of life [37], cervical exam participation and results (8.3%) [36], and body mass index (BMI) impact (8.3%) [32]. Lastly, analysis of gameplay was examined in two studies (16.7%) [32,38].

## Discussion

This study is the first to investigate the current state of research on VSGs designed for women's health education. The findings provide valuable insights into the populations targeted by the VSGs and the nature of these games. The limited number of studies examining VSG interventions, specifically in the development, applications, and effectiveness, indicates this area of research is still emerging. Eleven of the 12 studies were published in the last 10 years, suggesting a growing interest in VSGs for women's health education, potentially driven by advancements in technology and recognition of the potential benefits of VSGs. The geographic distribution of studies, i.e., the United States, Nepal, South Korea, and Norway highlights a global interest in utilizing VSGs for women's health and the need for further research from diverse regions.

In our exploration of which specific populations of women are targeted by VSGs in current research, we found that although only twelve studies were identified for this scoping review, the studies demonstrated diversity in the demographics of women studied. These demographics included low-income women and visible minority groups. Moreover, the studies covered a wide range of health education foci, addressing both general and specific women's health and health promotion issues. Further research examining the use and effectiveness of VSGs to educate women about health conditions is needed.

Our study revealed several key findings related to the characteristics of VSGs designed for women's health education. Our scoping review found that only about half of the studies on VSG interventions were grounded in established media, technology, and behavioural theories and frameworks [31–33,35,38,39]. However, existing literature highlights the significant role that educational theories and frameworks play in guiding the design and evaluation of VSGs in healthcare education [42]. Systematic reviews have further emphasized the need for a stronger theoretical perspective when designing and evaluating VSGs in healthcare professional education [43,44].

Various educational theories, such as the National League for Nursing/Jefferies Simulation Framework, Kolb's Experiential Learning Theory, and Bandura's Social Cognitive Theory and Concept of Self-Efficacy have been identified as applicable to simulation [45]. However, their integration into VSG development intended for patient education remains underexplored. To understand the design and effectiveness of VSGs, it is essential to consider how both educational design theories and game design frameworks are applied. While educational theories provide valuable insights into the learning process, they do not address the specific elements of game design that drive engagement and learning. Game design frameworks, such as Schell's Elemental Tetrad [46] and Winn's Design, Play, and Experience (DPE) Framework [47], are essential for understanding how game mechanics and dynamics work together to create an engaging gameplay experience that enhances learning. The success of VSGs in meeting learning objectives and enhancing self-efficacy depends on their ability to engage and motivate players. By considering the interplay between educational theories and game design frameworks, researchers can gain a more comprehensive understanding of how VSGs can be designed and implemented to optimize learning outcomes.

Although VSGs and simulations are often used interchangeably, they are not the same. VSGs are a specific type of simulation that incorporates game design elements to create interactive learning experiences. Despite this distinction, the reviewed studies did not address the use of instructional best practices or standardized simulation design frameworks specifically tailored for VSGs. Currently, the only identified set of standards identified in the literature is the Healthcare Simulation Standards of Best Practice (HSSOBP) developed by the International Nursing Association for Clinical Simulation and Learning (INACSL) [48–52]. However, these standards are primarily focused on general healthcare simulations and may not fully address the unique aspects of VSGs. Furthermore, there are no specific guidelines for using VSGs for patient education, indicating a gap in the literature. Although frameworks exist, such as those proposed by the National League for Nursing/Jeffries Simulation Frameworks [53,54], including a serious game development framework [10], and a simulation design template [55], future research should focus on incorporating standardized design criteria and established frameworks to enhance VSG development for patient or public education.

Serious game development frameworks exist, but they are not commonly used in designing virtual games specifically focused on women's health. Serious game frameworks offer structured approaches for creating games that combine learning objectives, instructional design, and game mechanics to promote education, training, or behaviour change. Frameworks such as the "Four-Dimensional Framework" [56] and the "Mechanics, Dynamics, and Anesthetics (MDA) Framework" [57] offer structured approaches that could be adapted to VSGs in this context. Additionally, integrating health behaviour change theories, such as the Health Belief Model or the Transtheoretical Model, could inform game design decisions that promote healthy behaviours and self-efficacy in women's health VSGs.

When designing VSGs for women's health, it is important to involve healthcare professional, women's health experts, and the target audience in the development process. This collaboration ensures that the game content is accurate, relevant, and addresses the unique health concerns and perspectives of women. Additionally, designers must navigate the challenges of presenting sensitive health information in an engaging and accessible manner while avoiding stigmatization or oversimplification. They must also consider the diverse experiences and backgrounds of women, ensuring that the game is inclusive and culturally sensitive.

The engagement of women from the target group in VSG development varied across studies. Five studies (41.7%) demonstrated engagement through community participation and user-centered co-design [31,33,34,38,39]. This approach supports the VSGs relevance, appropriateness, ease of use, and user understanding, ultimately enhancing the development of VSG [48]. The study's findings align with INACSL standards, emphasizing the need for a comprehensive needs assessment [48]. The needs assessment outlined in the standards includes a variety of analysis methods, such as analyzing the underlying causes of concern through root cause or gap analysis, conducting an organizational analysis, and surveying stakeholders, learners, clinicians, and educators to gather valuable input [48].

Similarly, collaboration and partnerships in VSG development among diverse stakeholders were also observed as key characteristics in the studies [31–35,38,39]. While some studies did not report collaborations or partnerships, others mentioned involvement from academic researchers [32,34], industry partners [34,35], oncology clinicians and advocacy organizations [31,33,38,39]. These collaborations foster an interdisciplinary approach and facilitate the exchange of knowledge, which is essential for developing high-quality VSGs. According to INACSL standards, simulations should be designed with content experts and simulationists who deeply understand best practices in simulation education, pedagogy, and practice [48]. Aligning with these standards and involving content experts in the development process can further enhance the effectiveness and quality of VSGs.

The reviewed studies employed diverse gaming modalities, including tablet-based [29,31,33,38,39], mobile-based [34,36,37], and web-based gaming [30,32,35,40]. However, it is essential to acknowledge the importance of aligning the chosen gaming modality with the learning objectives of the VSG, as recommended by the International Nursing Association for Clinical Simulation and Learning (INACSL) standards [48]. By considering the specific learning objectives and aligning the gaming modality, researchers and developers can enhance the VSGs educational value and relevance, ultimately contributing to improved learning outcomes and user experiences.

The gameplay dosage varied widely across studies, ranging from self-directed [31,32,36] or single-session play [29,30,33–35,40] to specific frequencies [37–39]. Gameplay duration (i.e., length) varied from self-directed [31,32,36,38–40] to specific durations [29,30,33–35,37], typically around 30 minutes [30,34,35]. Post-test timings also varied, with some assessments conducted immediately [29,30,34,40] and others scheduled at different intervals [32,35]. To the best of our knowledge, there is currently a lack of existing guidelines or standardized recommendations regarding the optimal dosage and timing of knowledge post-test after completing the VSGs. The observed variability across studies highlights the need for further research and the development of guidelines in this area. Future studies should specifically investigate the optimal timing of post-tests to assess long-term effects, contributing to advancing VSG interventions for health education.

In response to our research question regarding the outcomes examined in studies evaluating VSGs designed for women's health, it is evident that key insights emerge. The focus on knowledge outcomes, covering nutritional knowledge

[32,35], HIV and STIs awareness [29], breast cancer risk or awareness [30,40], and maternal and child health knowledge [34], aligns with existing literature emphasizing the importance of health literacy [58,59]. The assessment of self-advocacy skills [38,39], suggests a potential avenue for empowering women in their healthcare. The prominence of acceptability, likability, or satisfaction in 60% of studies [30,31,33–35,37,39], underscores the importance of the user experience, in line with the growing recognition of user-centered design in digital health interventions [48]. Health-related outcomes, encompassing behaviours [29,32], medication compliance, chemotherapy side effects, quality of life [37], and BMI impact [32] contribute to the broader discourse on the multifaceted nature of women's health. The inclusion of gameplay data analysis [32,38] provides a unique perspective, adding to discussions on technology integration in digital health interventions. The findings collectively emphasize the need for tailored, comprehensive approaches in developing and implementing VSGs for women's health.

### Strengths and limitations

This scoping review was limited to published English language studies and systematic or scoping reviews, excluding grey literature such as conference abstracts text, opinion papers, dissertations, and other unpublished material. This exclusion introduces potential bias, as the findings might not fully represent the breadth of available evidence. Additionally, the review excluded articles published in languages other than English, which could result in language bias and omitting valuable studies conducted in other languages.

Despite these limitations, the review demonstrates several strengths. One notable strength is the rigorous and transparent approach adopted for selecting relevant articles, extracting the data, and reporting the review. Adopting the JBI methodology [22] and adherence to the PRISMA-ScR guidelines [23] ensured a systematic and comprehensive review process, enhancing our study's overall reliability and replicability.

## Conclusions

In conclusion, this scoping review provides valuable insights into the current state of research on VSGs in women's health education. Over the last decade, there has been an increase in global interest regarding the use of VSGs for women's health education. This scoping review identified VSGs studies that target a range of age groups and were undertaken on three continents, i.e., North America, Asia, and Europe.

This scoping review explored the characteristics of VSGs, highlighting their application in various areas that target women's health: the incorporation of multiple theoretical perspectives, the importance of collaborative approaches with the target population, and partnerships with content experts and developers. Additionally, the scoping review examined design characteristics regarding gameplay dosage and outcomes, revealing various findings upon examination.

Due to the limited research available, it is essential to conduct future studies exploring the use of VSGs with other health conditions in the field of women's health. Furthermore, investigating the long-term effects and efficacy of VSGs in women's health education would contribute to a deeper understanding. By addressing these identified gaps, future studies can further advance the field of VSGs in women's health education.

### Implications for research

Several implications for future research should be considered to advance the development of VSGs in women's health education. Firstly, incorporating established educational theories and frameworks can enhance the design of VSGs, thereby optimizing the learning experience. In addition, integrating VSG design frameworks is suggested to improve the development of VSGs. Furthermore, involving women from the target population through community participation and a user-centered co-design process can enhance the relevance of VSGs. Fostering interdisciplinary collaborations and partners can facilitate knowledge exchange and enhance VSGs. It is also important to align gaming modalities with learning objectives and investigate optimal timing and game dosage. By exploring these areas, research can improve the

educational outcomes for women's health. These suggestions aim to further drive research and advancement in VSG interventions for women's health education.

## Supporting information

**S1 File.  Search strategy.**
(PDF)

**S2 File.  Studies ineligible following full-text review.**
(PDF)

**S3 File.  Data extraction tool.**
(PDF)

**S4 File.  Preferred Reporting Items for Systematic reviews and Meta-Analyses extension for Scoping Reviews (PRISMA-ScR) Checklist.**
(PDF)

## Acknowledgments

We extend our gratitude to Ilo-Katryn Maimets, the academic librarian, for her invaluable assistance in refining the search strategy.

## Author contributions

**Conceptualization:** Krissy Jordan, Celina Da Silva, Mahdieh Dastjerdi, Bella Grunfeld.

**Formal analysis:** Krissy Jordan.

**Methodology:** Krissy Jordan.

**Validation:** Krissy Jordan, Bella Grunfeld.

**Writing – original draft:** Krissy Jordan.

**Writing – review & editing:** Krissy Jordan, Christine Kurtz Landy, Celina Da Silva, Mahdieh Dastjerdi, Bella Grunfeld.

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
