## [Decision Letter · Decision Letter 0]

2 Apr 2025

PONE-D-24-37724Virtual Serious Games for Women’s Health Education: A Scoping ReviewPLOS ONE

Dear Dr. Jordan, 

Thank you for submitting your manuscript to PLOS ONE. After careful consideration, we feel that it has merit but does not fully meet PLOS ONE’s publication criteria as it currently stands. Therefore, we invite you to submit a revised version of the manuscript that addresses the points raised during the review process.

We look forward to receiving your revised manuscript.

Kind regards,

Raquel Inocencio da Luz, Phd

Academic Editor

PLOS ONE

Journal Requirements:

3. Please include a separate caption for figure 1 in your manuscript. 

Reviewers' comments:

Reviewer's Responses to Questions

**Comments to the Author**

1. Is the manuscript technically sound, and do the data support the conclusions?

Reviewer #1: Yes

2. Has the statistical analysis been performed appropriately and rigorously? 

Reviewer #1: N/A

3. Have the authors made all data underlying the findings in their manuscript fully available?

Reviewer #1: No

4. Is the manuscript presented in an intelligible fashion and written in standard English?

Reviewer #1: Yes

5. Review Comments to the Author

Reviewer #1: Thank you for the opportunity to review this paper! My area of expertise is in knowledge synthesis methodology, specifically search methodology, so I will be limiting the bulk of my feedback to methods, as I am not a subject expert.

Data availability

What you have included is great, but I wouldn't necessarily say this includes all data. This paper provides a great overview and recommendations for data availability for reviews: https://doi.org/10.1016/j.jclinepi.2022.03.003

I think it's mentioned in the paper but in case it's not, you are likely prohibited from sharing your actual export files from the databases because of license restrictions, everything else though is fair game!

Abstract

While I definitely appreciate you listing all the databases, your methods section in the abstract is otherwise a little scant. Please consult PRISMA-Abstract for a checklist of the kinds of things you should try to report in this section (I know it's intended for systematic reviews but where applicable it's still helpful for scoping reviews).

Line 106 - this is written a bit ambiguously. I assume you mean that you did not restrict by health condition or treatment?

You do a great job of differentiating between virtual simulations, VSGs and gamification, kudos!

Line 132 - Given the paucity of literature on the topic, and scoping reviews' general inclusion of more diverse information sources, I'm curious about the choice to exclude conference abstracts and disseratations/theses in particular, as I could see there being rich content here. Not saying you're wrong for excluding them but I do think you need some strong rationale here for why not.

Not sure if this is a PLoS One thing but I'm surprised that all your inclusion criteria (basically lines 103-133) are in the Introduction section rather than in the Methods. Maybe double check this!

Line 136 Great distinction between conduct and reporting guidelines, kudos!

Line 137 You mention a protocol but not where it can be accessed. Is it on OSF or another repository?

Line 146 Your searches are great and I loved that you reported all of them well done! I see that the librarian is acknowledge in your paper, that's also great.

Line 161 Was full-text also screened in duplicate?

Line 209 It may be helpful here to differentiate between studies and reports. e.g. One study was described in four separate reports. As you get into Characteristics of included studies this will help reduce confusion because I'm unsure here if you're quadruple counting that one study or only counting it the once (or if in some cases you're describing reports).

Line 218 It seems like you are double counting here as all of these add up to 12, whereas I would think it should count up to 9, right? (8 individual studies plus the 9th study that has 4 reports/pubs associated with it?). I think you need to go through this section and make sure you're not inadvertently weighting things incorrectly by including all four reports of that one study, and decide where it makes sense to report them as a big glob or separately.

Final thoughts

This is a very well conducted and reported review. Some things to iron out but I think it's a great paper and will be a great addition to the literature.

6. PLOS authors have the option to publish the peer review history of their article (what does this mean? ). If published, this will include your full peer review and any attached files.

**Do you want your identity to be public for this peer review?** For information about this choice, including consent withdrawal, please see our Privacy Policy .

Reviewer #1: No

---

## [Author Response · Author response to Decision Letter 1]

16 Apr 2025

Response to Reviewers

Journal Requirements:

Thank you for your feedback. We have revised the manuscript to meet PLOS ONE’s style requirements. Additionally, we have updated the supplementary files.

Thank you for your message and for the opportunity to clarify our data-sharing plan.

The data analyzed in this scoping review consists of third-party, peer-reviewed articles retrieved from licensed databases, including MEDLINE, CINAHL, EMBASE, Web of Science, and PsycINFO. Due to copyright and licensing restrictions, we are unable to share full-text articles or database export files, as doing so would violate the terms of use set by the database providers.

However, in alignment with PLOS ONE’s open data policy and to ensure transparency and reproducibility, we have made all other data generated during the review process fully available in the Supporting Information files, which include:

• S1 File: Full search strategy

• S2 File: Studies ineligible following full-text review

• S3 File: Data extraction tool

These materials are provided in accessible formats to support reuse in accordance with FAIR data principles.

All included studies are fully cited within the manuscript, and upon acceptance, the study protocol will be publicly available on protocols.io at DOI: dx.doi.org/10.17504/protocols.io.j8nlkd4z5g5r/v1

We respectfully request an exemption from sharing full-text articles and database export files due to these licensing restrictions. We hope this approach satisfies the journal’s data policy requirements, and we are happy to revise further if needed.

3. Please include a separate caption for figure 1 in your manuscript.

We have added a separate caption for figure 1 in the manuscript.

To the best of our knowledge, none of the cited papers have been retracted, and all references are current.

Reviewers' comments:

Reviewer's Responses to Questions

Comments to the Author

1. Is the manuscript technically sound, and do the data support the conclusions?

Reviewer #1: Yes

2. Has the statistical analysis been performed appropriately and rigorously?

Reviewer #1: N/A

3. Have the authors made all data underlying the findings in their manuscript fully available?

Reviewer #1: No

This manuscript is a scoping review based on previously published, peer-reviewed literature. As confirmed by the journal, it is not necessary to share the full-text articles analyzed, provided that all included studies are properly cited in the manuscript. Due to copyright and database licensing restrictions, we cannot share full-text articles or database exports.

In alignment with PLOS ONE’s data policy and based on the editorial guidance received via emails, we have updated the Data Availability Statement in the submission system and manuscript to read:

“All relevant data are within the article and its Supporting Information files.”

To promote transparency and reproducibility, all data generated during the review process are included in the Supporting Information, specifically:

• S1 File: Full search strategy

• S2 File: List of studies excluded after full-text review

• S3 File: Data extraction tool

This approach ensures compliance with PLOS ONE’s open data requirements while respecting third-party content restrictions.

We hope this revision meets the reviewer's expectations and aligns with the journal’s data policy.

4. Is the manuscript presented in an intelligible fashion and written in standard English?

Reviewer #1: Yes

5. Review Comments to the Author

Reviewer #1: Thank you for the opportunity to review this paper! My area of expertise is in knowledge synthesis methodology, specifically search methodology, so I will be limiting the bulk of my feedback to methods, as I am not a subject expert.

1. Data availability - What you have included is great, but I wouldn't necessarily say this includes all data. This paper provides a great overview and recommendations for data availability for reviews: https://doi.org/10.1016/j.jclinepi.2022.03.003

I think it's mentioned in the paper but in case it's not, you are likely prohibited from sharing your actual export files from the databases because of license restrictions, everything else though is fair game!

Thank you for your thoughtful feedback and for sharing the helpful reference. We appreciate the emphasis on improving transparency regarding data availability for reviews.

We acknowledge that, while full-text articles and raw database export files cannot be shared due to licensing restrictions, the remaining components of our data—including our search strategy, data extraction tool, and a list of studies excluded after full-text review—are fully available in the Supporting Information files.

In line with the recommendations in the cited paper, we have reviewed our materials to ensure that all data generated during the review process (and permissible to share) are included. We have also updated the Data Availability Statement to reflect the journal's guidance:

“All relevant data are within the article and its Supporting Information files.”

2. Abstract - While I definitely appreciate you listing all the databases, your methods section in the abstract is otherwise a little scant. Please consult PRISMA-Abstract for a checklist of the kinds of things you should try to report in this section (I know it's intended for systematic reviews but where applicable it's still helpful for scoping reviews).

Thank you for the suggestion. We have consulted the PRISMA-Abstract checklist for scoping reviews and revised the methods section of the abstract to improve its clarity and completeness.

3. Line 106 - this is written a bit ambiguously. I assume you mean that you did not restrict by health condition or treatment?

Thank you for this helpful clarification. You are correct; we did not impose restrictions based on participants’ health conditions or treatment statuses. To clarify, we have revised the sentence as follows:

“This review considered studies exclusively involving participants identifying as women or females aged 18 years and older, with no restrictions based on health condition or treatment status.”

4. You do a great job of differentiating between virtual simulations, VSGs and gamification, kudos!

Thank you very much – we appreciate your positive feedback!

5. Line 132 - Given the paucity of literature on the topic, and scoping reviews' general inclusion of more diverse information sources, I'm curious about the choice to exclude conference abstracts and disseratations/theses in particular, as I could see there being rich content here. Not saying you're wrong for excluding them but I do think you need some strong rationale here for why not.

Thank you for this insightful comment. We agree that grey literature can offer valuable perspectives, particularly in emerging areas of research. However, we chose to exclude grey literature (e.g., conference abstracts and dissertations) to maintain consistency in methodological rigour and reporting quality across sources. These types of sources often lack sufficient detail to support reliable data extraction. We have now added this rationale to the manuscript to clarify our approach.

6. Not sure if this is a PLoS One thing but I'm surprised that all your inclusion criteria (basically lines 103-133) are in the Introduction section rather than in the Methods. Maybe double check this!

Thank you for highlighting this. We agree that the inclusion criteria are more appropriately placed in the Methods section. We have moved this content (previously lines 103-133) to a new “Inclusion criteria” subheading within the Methods section to enhance clarity and manuscript structure.

7. Line 136 - Great distinction between conduct and reporting guidelines, kudos!

Thank you for the kind words. We’re glad this distinction clearly came through.

8. Line 137 - You mention a protocol but not where it can be accessed. Is it on OSF or another repository?

Thank you for catching this. We have updated the manuscript to indicate that the protocol is publicly available via protocols.io. The protocol will be accessible at DOI: dx.doi.org/10.17504/protocols.io.j8nlkd4z5g5r/v1 upon publication. In the meantime, reviewers can access it via this private link: https://www.protocols.io/private/D487D76E193B11F098870A58A9FEAC02

9. Line 146 - Your searches are great and I loved that you reported all of them well done! I see that the librarian is acknowledge in your paper, that's also great.

Thank you very much! We appreciate your recognition of the thoroughness of our search strategy and the librarian’s contribution. Their support was invaluable in ensuring the comprehensiveness of the searches.

10. Line 161- Was full-text also screened in duplicate?

Thank you for your question. Yes, full-text screening was conducted independently by two reviewers, with discrepancies resolved through discussion. We have clarified this in the manuscript.

11. Line 209 - It may be helpful here to differentiate between studies and reports. e.g. One study was described in four separate reports. As you get into Characteristics of included studies this will help reduce confusion because I'm unsure here if you're quadruple counting that one study or only counting it the once (or if in some cases you're describing reports).

Thank you for your important suggestion. We have clarified that four distinct studies examined the same VSG intervention but reported different outcomes. This has been addressed in the revised manuscript to avoid confusion and ensure transparency in reporting.

12. Line 218 - It seems like you are double counting here as all of these add up to 12, whereas I would think it should count up to 9, right? (8 individual studies plus the 9th study that has 4 reports/pubs associated with it?). I think you need to go through this section and make sure you're not inadvertently weighting things incorrectly by including all four reports of that one study, and decide where it makes sense to report them as a big glob or separately.

Thank you for your thoughtful feedback. We appreciate your concern regarding potential double counting. To clarify, four published studies related to the same VSG intervention addressed different research questions and outcomes. Since our objective is to map the current state of research on VSGs for women’s health – not to quantify interventions- we have reported them as individual studies. We reviewed this section to ensure accurate counting and have revised the text to clarify that these studies stem from the same intervention but represent separate analyses. This clarification builds on the revision made in response to comment #11.

13. Final thoughts - This is a very well conducted and reported review. Some things to iron out but I think it's a great paper and will be a great addition to the literature.

Thank you for your thoughtful and encouraging feedback. We truly appreciate your detailed review and have worked diligently to address all of your suggestions to enhance our scoping review.

6. PLOS authors have the option to publish the peer review history of their article (what does this mean?). If published, this will include your full peer review and any attached files.

Do you want your identity to be public for this peer review? For information about this choice, including consent withdrawal, please see our Privacy Policy.

Reviewer #1: No

---

## [Decision Letter · Decision Letter 1]

12 May 2025

Virtual Serious Games for Women’s Health Education: A Scoping Review

PONE-D-24-37724R1

Dear Authors,

We’re pleased to inform you that your manuscript has been judged scientifically suitable for publication and will be formally accepted for publication once it meets all outstanding technical requirements.

Kind regards,

Raquel Inocencio da Luz, Phd

Academic Editor

PLOS ONE

Additional Editor Comments (optional):

Reviewers' comments:

Reviewer's Responses to Questions

**Comments to the Author**

1. If the authors have adequately addressed your comments raised in a previous round of review and you feel that this manuscript is now acceptable for publication, you may indicate that here to bypass the “Comments to the Author” section, enter your conflict of interest statement in the “Confidential to Editor” section, and submit your "Accept" recommendation.

Reviewer #1: All comments have been addressed

2. Is the manuscript technically sound, and do the data support the conclusions?

Reviewer #1: Yes

3. Has the statistical analysis been performed appropriately and rigorously? 

Reviewer #1: N/A

4. Have the authors made all data underlying the findings in their manuscript fully available?

Reviewer #1: Yes

5. Is the manuscript presented in an intelligible fashion and written in standard English?

Reviewer #1: Yes

6. Review Comments to the Author

Reviewer #1: Thank you for your revisions, you've addressed all of my feedback!

7. PLOS authors have the option to publish the peer review history of their article (what does this mean? ). If published, this will include your full peer review and any attached files.

**Do you want your identity to be public for this peer review?** For information about this choice, including consent withdrawal, please see our Privacy Policy .

Reviewer #1: No

---

## [Editor Report · Acceptance letter]

PONE-D-24-37724R1

PLOS ONE

Dear Dr. Jordan,

I'm pleased to inform you that your manuscript has been deemed suitable for publication in PLOS ONE. Congratulations! Your manuscript is now being handed over to our production team.

Kind regards,

on behalf of

Dr Raquel Inocencio da Luz

Academic Editor

PLOS ONE